# Urinary Excretion of Iohexol as a Permeability Marker in a Mouse Model of Intestinal Inflammation: Time Course, Performance and Welfare Considerations

**DOI:** 10.3390/ani11010079

**Published:** 2021-01-04

**Authors:** Victoria Ortín-Piqueras, Tobias L Freitag, Leif C Andersson, Sanna H Lehtonen, Seppo K Meri, Thomas Spillmann, Rafael Frias

**Affiliations:** 1Department of Equine and Small Animal Medicine, Faculty of Veterinary Medicine, University of Helsinki, FIN-00014 Helsinki, Finland; thomas.spillmann@helsinki.fi; 2Comparative Medicine, Karolinska Institute, SE-171 77 Stockholm, Sweden; rafael.frias@ki.se; 3Translational Immunology Research Program, University of Helsinki, FIN-00014 Helsinki, Finland; tobias.freitag@helsinki.fi (T.L.F.); seppo.meri@helsinki.fi (S.K.M.); 4Department of Pathology, University of Helsinki, FIN-00014 Helsinki, Finland; leif.andersson@helsinki.fi (L.C.A.); sanna.h.lehtonen@helsinki.fi (S.H.L.); 5Research Programme for Clinical and Molecular Medicine, University of Helsinki, FIN-00014 Helsinki, Finland

**Keywords:** duodenitis, inflammation, intestinal permeability, iohexol, metabolic cage, mouse, permeability testing, small intestine

## Abstract

**Simple Summary:**

In mammals, different diseases are associated with intestinal changes that may cause an increase in gut permeability. Intestinal permeability tests allow the evaluation of intestinal damage in humans, veterinary patients and laboratory animal models. When used in mouse models, these tests require that animals are singly housed in metabolic cages with a wire-grid floor to collect urine samples. This raises welfare concerns. Iohexol meets several criteria for an ideal intestinal permeability marker and has recently been used in several species. Here, we examined the performance of an intestinal permeability test using iohexol administered by mouth and following excretion over 24 h in urine. As a model, we chose immunodeficient mice with intestinal inflammation induced by adoptive transfer of effector/memory T cells. We collected urine samples at seven time points to profile the urinary excretion of iohexol, in addition to intestinal tissue samples for histological assessment. We conclude that a 6 h cumulative urine sample may be sufficient to evaluate small intestinal permeability in this mouse model and increased urinary excretion of iohexol is correlated with increased severity of duodenitis. The welfare of mice housed in metabolic cages could be improved by reducing the cage periods from 24 to 6 h.

**Abstract:**

Intestinal permeability (IP) tests are used to assess intestinal damage in patients and research models. The probe iohexol has shown advantages compared to ^51^Cr-EDTA or absorbable/nonabsorbable sugars. During IP tests, animals are housed in metabolic cages (MCs) to collect urine. We examined the performance of an iohexol IP test in mice. Rag1-/- (C57BL/6) mice of both sexes were divided into controls or treatment groups, the latter receiving injections of effector/memory T cells to induce intestinal inflammation. After two, four and five weeks (W), a single dose of iohexol was orally administered. Urine was collected seven times over 24 h in MCs. Iohexol concentration was measured by ELISA. Intestinal histological damage was scored in duodenal sections. In control and treated mice of both sexes, urinary excretion of iohexol peaked at 4 h. From W2 to W4/W5, urinary iohexol excretion increased in treated mice of both sexes, consistent with development of duodenitis in this model. Positive correlations were observed between the urinary excretion of iohexol in W4/W5 and the histological severity of duodenitis in treated male mice. We conclude that a 6 h cumulative urine sample appears sufficient to evaluate small IP to iohexol in this mouse model, improving animal welfare by reducing cage periods.

## 1. Introduction

Intestinal permeability (IP) tests are valuable diagnostic tools to screen, monitor and evaluate hyperpermeability and the degree of damage to the intestinal barrier in humans, veterinary patients and experimental in vivo models [1,2,3,4,5,6]. However, animal models and standardized IP testing protocols still need improvement, especially to further investigate disease etiology and pathophysiology, test the effect of therapeutic agents and reduce data variation across studies and gain consistency in the probes used [4,7,8]. Challenges in the standardization of IP tests concern the choice of markers, variations in the procedure and the method of sampling [9].

Chromium 51-labeled ethylenediaminetetraacetic acid (^51^Cr-EDTA) and/or a combination of mono- and disaccharides (e.g., mannitol and lactulose) have been defined as the gold standard to test IP over the past years [10]. Disadvantages mainly associated with the radioactivity of ^51^Cr-EDTA and the bacterial degradation of the sugars have led to the search for more appropriate candidates [4,10,11]. Iohexol, a nonradioactive, stable, water-soluble, radiographic contrast medium, has been successfully used as an IP marker in humans and various animal species [3,12,13,14,15,16,17,18,19,20]. Iohexol is not metabolized, resulting in superior stability compared to other agents [19]. Samples containing this contrast agent may be safely (re-)analyzed if required without any apparent degradation of iohexol [21,22]. The concentration of iohexol in very small samples of both urine and blood has been successfully measured by using an enzyme-linked immunosorbent assay (ELISA) in different species [2,3,16,23].

Intestinal permeability testing has not yet been widely applied in mice [8]. Serial blood sampling for longitudinal studies in mice is invasive and subject to practical limitations. Thus far, IP methods have more frequently relied on single or terminal blood sampling during the final stage of studies [24]. Most IP tests are considered less invasive if mouse urine is used instead. This allows the reuse and retesting of animals throughout a longitudinal study. More recently, an iohexol IP test has been adapted for use in a coeliac disease mouse model [23]. However, the use of specific equipment such as metabolic cages (MCs) is required to accurately collect urine specimens, usually during a 24 h (h) period [4]. Mice must be singly housed on wire-mesh floors without bedding or enrichment [25,26,27,28]. These isolating cages are known to induce animal stress, which may lead to significant behavioral and physiological changes and compromise animal welfare. Mice have been shown to adapt poorly to MCs, requiring three to four days to reach equilibrium [29,30]. For all these reasons combined, IP tests in mice are rarely used in practice.

The overall objective of this study was to (1) analyze the performance of an iohexol IP test in a mouse model of small intestinal and colonic inflammation [31], (2) determine the time course of urinary excretion of iohexol in mice and (3) optimize the procedure from an animal welfare perspective. Our main hypothesis was that animals with intestinal inflammatory damage have increased IP to iohexol compared to treatment-naïve mice and that IP increases with the development of duodenitis over time. Another hypothesis was that IP might be assessed within a shorter period than the commonly used 24 h MC housing after the oral administration of iohexol.

## 2. Materials and Methods

### 2.1. Study Approval

The study was planned according to ARRIVE guidelines and was approved by the animal research board of the Southern Finnish State Administrative Agency (ESAVI/1286/04.10.07/2016 and ESAVI/8249/04.10.07/2015).

### 2.2. Animals and Housing

The experiments were performed in Rag1-/- recipient mice (C57BL/6J genetic background; 12–14 weeks of age; males and females). C57BL/6J wildtype mice were used as T cell donors (10–24 weeks of age; males and females). All mice were bred and raised under specific pathogen-free (SPF) conditions in colonies maintained at the Laboratory Animal Centre (LAC) on the Viikki campus of the University of Helsinki, Finland. Breeders had originally been purchased from the Jackson Laboratory (Bar Harbor, ME, USA). New animals are only introduced into this unit by rederivation or from certified commercial breeders. Experimental mice were transferred to a conventional unit on the Meilahti campus of the university to conduct all the experiments. The health of both units is regularly monitored by undertaking annual tests for all FELASA-listed pathogens [32,33] as well as quarterly serological and bacteriological tests. While the facility on the Viikki campus was negative for all tested pathogens, mouse norovirus, *Staphylococcus aureus* (by culture, no lesions) and Helicobacter spp. (not *H. bilis*, *H.hepaticus* or *H. rodentium*) were identified in sentinel animals of the conventional unit on the Meilahti campus during routine health monitoring.

All mice were kept under a 14:10 h light–dark cycle (lights on at 06:00 and off at 20:00) at 22 ± 2 °C and 50–60% humidity. Animals had ad libitum access to either a gluten-free standard diet (AIN-76A) or a similar diet containing 2.5 g of wheat gluten (Sigma) per kg (modified from AIN-76A; both diets produced and irradiation-sterilized at Research Diets, New Brunswick, NJ, USA). Filtered tap water was provided without restrictions in polycarbonate bottles, and the cages were cleaned once per week. Mice were housed in individually ventilated plastic cages (Mouse IVC Green Line, 391 × 199 × 160 mm, floor area 501 cm^2^; Tecniplast, Buguggiate, Italy), with half of the cage covered by a wire bar food hopper. Aspen wood chips were provided as bedding (5 × 5 × 1 mm; Tapvei Oy, Kaavi, Finland), along with red polycarbonate mouse igloos and Aspen wood wools (Datesand Group and Tapvei Oy) for environmental enrichment and nesting material. A maximum of five mice were housed per cage.

Animals were weighed twice weekly by the same experimenter at the same time over the length of the study. Stool consistency and appearance were visually monitored for the duration of the study and the presence of watery feces or diarrhea was noted. The animal physical examination consisted of visual inspection of the animals’ posture, appearance and spontaneous behavior, including symptoms of potential pain and/or distress, e.g., piloerection, an unkempt coat/rough hair, a hunched posture and apathy [34,35]. Mice were routinely handled during the procedures by using a tunnel or cupped hand to avoid picking them up by the base of the tail. Tail handling has been shown to induce aversion and high anxiety levels that can compromise animal welfare and affect the scientific outcomes [36].

### 2.3. Experimental Design and Procedures

#### 2.3.1. Time Points of Iohexol Measures

Thirty-five (17♀, 18♂) Rag1-/- mice, ranging in body weight from 24.3 to 30.6 g in males and 16.4 to 20.9 g in females, were used in this study. Eleven mice (5♀, 6♂) were allocated to the control group and the remaining twenty-four mice (12♀, 12♂) to the treatment group. The sample size calculation was based on previous publications using iohexol as a marker of IP [1,3,16,23].

After the induction of intestinal inflammation in treated mice (Week 0; Section 2.3.2), a total of eight IP tests were performed over seven consecutive weeks (Figure 1). All mice were acclimatized one week before beginning the T cell administration (Week 1). In Weeks 1 and 7 (W1 and W7, i.e., at the beginning and end of the IP study), all the untreated control female and male mice (*n* = 11) were IP tested. In Weeks 2, 4 and 5 (W2, W4 and W5), the treated mice (*n* = 12; *n* = 11 in W5) were all tested. The recovery period between the IP test in W2 and W4 was 14 days, while that between W4 and W5 was 7 days. For the control group, the recovery period between the IP test in W1 and W7 was 6 weeks. During W5 (treated mice) and W7 (control mice), all mice were euthanized. The mice were exsanguinated under ketamine–xylazine anesthesia, and tissue samples were collected for histological assessment (see below).

#### 2.3.2. Induction of intestinal Inflammation

Intestinal inflammation was induced in Rag1-/- recipients by intraperitoneal injection of 4.5 × 10^5^ fractionated, sex-matched, splenic CD4+CD62L-CD44 high effector/memory T cells from C57BL/6J donors in 400 µL of PBS (treated Rag1-/- mice; Figure 1) [31,37]. This model is characterized by T-cell-mediated duodenitis, enteritis and colitis. Typically, treated Rag1-/- mice present with weight loss and diarrhea, starting 4 weeks from the injection of T cells.

#### 2.3.3. Treatment of Adoptive T Cell Recipient Mice with Different Diets

Immediately following adoptive T cell transfer, two subgroups of treated mice received either a gluten-free (*n* = 6 female + 6 male mice) or gluten-containing (*n* = 6 female + 6 male mice) diet until the end of the study (for a diet composition, see Section 2.2). Our intention initially was to study differences in duodenitis severity induced by diet modification. Because our study remained underpowered to perform this comparison, based on previous experience [31,37,38], we decided to pool both treated subgroups for analysis. Below, we refer to both subgroups as “treated mice.” Untreated control mice were maintained on a gluten-free diet throughout the study.

#### 2.3.4. Measurement of Intestinal Permeability

The iohexol IP test consisted of a single dose of iohexol (10 mL/kg, 647.1 mg/mL, Omnipaque 300^®^, Amersham Health, The Netherlands) administered by orogastric gavage to each mouse using a malleable stainless steel feeding needle and a 1 mL syringe [23]. The recommended iohexol dosage for the assessment of intestinal permeability in mice was adjusted from Frias et al. [1]. After iohexol dosing, the animals were placed in individual MCs for urine collection for 24 h. Urine samples were taken seven times over 24 h (at 2, 4, 6, 12, 15, 18 and 24 h), starting when the lights turned off in the animal room (Figure 2). This moment is the beginning of the active phase of the circadian rhythm of the mice [39,40,41]. During this phase, a dim red light was used to maintain the circadian organization and to avoid any circadian disturbance in the animals during the experiment [42,43,44]. No sedative drug was used before, during or after iohexol administration. Animals were randomly oral-dosed and allocated to each MC by using a list randomizer program (https://www.random.org/lists/). Three food pellets and bottles of fresh tap water were provided before using the MCs.

The volume of recovered urine was individually recorded, and urine samples were kept in Eppendorf tubes and frozen at −20 °C. If fecal contamination of urine was observed, the affected sample was discarded.

#### 2.3.5. Iohexol Measurement

The iohexol concentration in mouse urine was determined by ELISA according to the manufacturer’s instructions (FIT–GFR^TM^ Iohexol Kit, BioPAL Inc., Worcester, MA, USA) and also as described by Ortín-Piqueras et al. [2]. Before analysis, urine samples were brought to room temperature and thoroughly mixed. All samples were analyzed in duplicate to increase the reliability of the analysis. The urinary excretion of iohexol (% of ingested iohexol) in mice was calculated using the following equation [1,3,16]:$$\mathrm{Iohexol}\;\mathrm{in}\;\mathrm{urine}\;\left(\%\right)\;=\;\frac{\mathrm{amount}\;\mathrm{of}\;\mathrm{iohexol}\;\mathrm{excreted}\;\mathrm{in}\;\mathrm{urine}\;\mathrm{after}\;\mathrm{each}\;\mathrm{time}\;\mathrm{period}\;\left(\mathrm{mg}\right)}{\mathrm{amount}\;\mathrm{of}\;\mathrm{administered}\;\mathrm{iohexol}\;\left(\mathrm{mg}\right)}\times \;100.\;(1)\;=\frac{\mathrm{concentration}\;\mathrm{of}\;\mathrm{iohexol}\;\mathrm{in}\;\mathrm{urine}\;\left(\frac{\mathrm{mg}}{\mathrm{mL}}\right)\times \;\mathrm{DF}\;\times \;\mathrm{TUV}\;(\mathrm{mL})}{\mathrm{volume}\;\mathrm{of}\;\mathrm{iohexol}\;\mathrm{given}\;\mathrm{orally}\;\left(\mathrm{mL}\right)\;x\;\mathrm{concentration}\;\mathrm{of}\;\mathrm{iohexol}\;\mathrm{solution}\;\left(\frac{\mathrm{mg}}{\mathrm{mL}}\right)}\times \;100\;\mathrm{DF},\;\mathrm{dilution}\;\mathrm{factor};\;\mathrm{TUV},\;\mathrm{total}\;\mathrm{urine}\;\mathrm{volume}\;\mathrm{recovered}\;\mathrm{within}\;\mathrm{certain}\;\mathrm{hours}$$

#### 2.3.6. Tissue Collection and Histological Score for Duodenitis

Mice were exsanguinated under ketamine/xylazine anesthesia after either 5 or 7 weeks (see Figure 1). The weight of the entire small intestine, duodenum (2 cm) and colon was recorded (weights including bowel content). For the histological assessment, duodenal samples were fixed in formalin and embedded in paraffin, and longitudinal sections (6 μm) were stained with hematoxylin and eosin (H/E). To identify proliferating cells in additional sections, the Lab Vision PT Module (Thermo Fisher, Waltham, MA, USA) was used. For antigen retrieval, sections were immersed in TRIS-HCl, pH 8.5, prewarmed to 65 °C for 30 min, and treated at 98 °C for 30 min. Sections were stained using polyclonal rabbit anti-mouse Ki-67 IgG (dilution 1:500, incubation at RT for 1 h; Bethyl Laboratories), peroxidase blocking (incubation 15 min), biotinylated goat anti-rabbit IgG, avidin–biotin complex reagent (incubation both for 30 min) and diaminobenzidine substrate (incubation for 10 min; all from Vector Laboratories). Hematoxylin (incubation for 45 s) was used for counterstaining. The histological score for duodenitis was assessed in a blinded fashion in well-oriented sections and at sites representative of maximal damage, as reported previously [38]. The length of 5 crypts (stained epithelial cells) and 5 villi (unstained epithelial cells) was measured in duodenal sections stained for Ki-67, and villus/crypt architecture scores were assigned as follows: score 0 (V/C ratio > 3.00), 0.5 (2.50–3.00), 1.0 (2.00–2.49), 1.5 (1.50–1.99), 2.0 (1.00–1.49), 2.5 (0.50–0.99), 3.0 (<0.5). Villus cellular infiltration was graded in H/E-sections as follows, assessing 5 villus/crypts units: score 0 (villus lamina propria diameter < 0.5 × crypt diameter), 1 (0.5–1×), 2 (1–2×), 3 (>2×). Basal infiltration with neutrophils was graded in H/E-sections as follows: score 0 (normal crypts), 1 (1–7 crypt abscess(s) per duodenal section), 2 (8–15 crypt abscesses), 3 (>15 crypt abscesses). Each specimen was assigned a composite histological score for duodenitis by combining the above three separate parameters (maximum score: 3 + 3 + 3 = 9).

### 2.4. Statistical Analysis

All statistical analyses were performed and graphs were drawn with SPSS (IBM SPSS 26.0, SPSS Inc., Chicago, IL, USA) and GraphPad Prism (GraphPad Prism 8, GraphPad Software Inc., San Diego, CA, USA), respectively. The model fit was assessed by evaluating studentized model residuals graphically (normal QQ plot) and with histograms and by a test of normality (Kolmogorov–Smirnov). Decimal logarithm (Log10 (1 + x)) transformation was used for the urinary excretion of iohexol to satisfy the normality assumption. Differences in the number of animals delivering urine, the volume of urine, animal body weight and urinary excretion of iohexol were analyzed using either the Mann–Whitney U-test or t-test to calculate statistical differences between two independent variables or Kruskal–Wallis or one-way ANOVA for more than two variables. Post hoc tests were applied for multiple comparisons with Bonferroni correction. A linear mixed model test was performed to analyze the interaction between fixed effects described as follows: group (control and treatment), urine analysis time point (2, 4, 6, 12, 15, 18 and 24 h), sex (female and male) or treatment effect analysis week (W1, W2, W4, W5 and W7) with two-way interactions of each independent variable. The results were expressed as the median ± interquartile range (±IQR) or mean ± standard deviation (±SD). All correlations were calculated by using Spearman’s rank correlation coefficient (r). Values of *p* < 0.05 were considered statistically significant.

## 3. Results

### 3.1. Clinical Examination

Body weight, fecal consistency and the health condition of the animals were monitored throughout the study. During each IP testing session, animal behavior was observed. Overall, the mice were in good physical condition and behaved normally. They displayed normal behavior after returning to their home cages from the MCs. Only one male mouse from the treated group was removed during the running IP test in W5. This animal displayed changes in behavior, i.e., apathy, rough hair coat and hunched posture, at the start of the experiment and was excluded from W5 iohexol analysis.

The stool consistency for all mice was mostly reported as normal. Occasional transient wet or pasty stools were observed in W4 or W5, respectively, in some female and male mice from the treatment group. These changes in stool consistency were only obvious by visual observation during MC housing. One urine sample was discarded from iohexol analysis due to fecal contamination.

The median ± IQR weight for control and treated female mice was 19.15 g ± 1.10 and 20.00 g ± 1.10, respectively, and for control and treated male mice it was 29.10 g ± 1.90 and 24.70 g ± 1.40, respectively. The weight curve for the entire study and the changes in body weight after each period of MC housing are presented in Appendix A. In both female and male mice, there was a significant difference in the median ± IQR weight between the control and treatment groups (*p* < 0.001). In control and treated male mice, there were some drops in body weight immediately after the IP test in MC housing without any significant difference (Appendix A).

### 3.2. Number of Animals Delivering Urine and Volume of Collected Urine

The median ± IQR number of control (*n* = 11) and treated mice (*n* = 12) that urinated during the cumulative seven time points over the 24 h period is illustrated in Appendix A. The median ± IQR number of control mice that urinated during the IP tests (cumulative hours) was 2.00 ± 0.00 (2 h), 8.50 ± 3.00 (4 h), 10.50 ± 1.00 (6 h) and 11.00 ± 0.00 (12–14 h), respectively (Appendix A). The median ± IQR number of treated mice was 3.50 ± 0.50 (2 h), 9.50 ± 0.50 (4 h) and 12.00 ± 0.50 (6–24 h), respectively (Appendix A). In total, the median ± IQR number of animals delivering urine was 3.00 ± 2.00 (2 h), 9.50 ± 2.00 (4 h), 11.50 ± 1.50 (6 h), 12.00 ± 1.00 (12 h), 12.00 ± 1.00 (15 h), 12.00 ± 1.00 (18 h) and 12.00 ± 1.00 (24 h), respectively. This represents 25% (2 h), 79.2% (4 h), 95.8% (6 h) and 100% (12–24 h) of all mice.

The total volume (µL) of mouse urine collected during the IP tests is presented in Appendix A and Appendix A. In treated and control mice, the volume of urine peaked at 4 h, followed by a less pronounced increase at 24 h. In control mice in W1, the first peak in urine volume was at 2 h. The median ± IQR volume of urine in control mice in W1 and W7, and in treated mice in W2, W4 and W5 was 108.00 ± 240.00, 48.00 ± 169.99, 60.00 ± 96.50, 55.00 ± 74.75 and 54.5 ± 77.00 µL, respectively. There was a significant difference in the volume of mouse urine collected during the IP tests between control and treated mice (*p* = 0.014), and more specifically between W1 and W2 (*p* = 0.022), W1 and W4 (*p* = 0.019) and W1 and W5 (*p* = 0.001).

### 3.3. Urinary Excretion of Iohexol during 24 h

The urinary excretion of iohexol was examined by determining the percentage of orally administered iohexol recovered in mouse urine (Figure 3, Appendix A). When combining the data from all measurement time points during the study, there was no significant difference in the urinary excretion of iohexol between all treated and control mice (*p* = 0.223). Data analysis revealed a significant difference in the urinary excretion of iohexol (%) between the treatment groups in W4 and W5 compared to that in W2 (*p* < 0.001) (Figure 3A). There was also a significant difference in the urinary excretion of iohexol when comparing both control mice (W1 and W7) and treated mice in W2 with treated mice in W4 and W5 (Figure 3B).

When combining the data from all measurement time points during the study, there was no significant difference in the urinary excretion of iohexol between male and female mice (*p* = 0.532). In female mice, there was a significant difference in the urinary excretion of iohexol between treated mice in W2 and W4 (*p* = 0.048) and W2 and W5 (*p* = 0.002) and between control mice in W7 and treated mice in W5 (*p* = 0.033) (Figure 3C, Appendix A). In male mice, a significant difference in the urinary excretion of iohexol occurred in treated mice between W2 and W4 (*p* < 0.001) as well as between W2 and W5 (*p* < 0.001) (Figure 3D, Appendix A).

In control and treated mice, there was a significant difference in the urinary excretion of iohexol between the seven urinary collection time points (*p* < 0.001). In both treated and control mice, the urinary excretion of iohexol peaked at 4 h. There was also a second and less pronounced increase at 24 h in control mice and in treated mice only when IP was tested in W4 and W5 (Figure 4A,B, Table 1 and Appendix A). These increases in the urinary excretion of iohexol at 4 h and 24 h were higher in the treated mice in W4 and W5 than in the treated mice in W2 and the control mice. At 4 and 24 h, the strongest significant differences in the urinary excretion of iohexol occurred between W2 and W4 (*p* = 0.003, *p* = 0.007, respectively) and W2 and W5 (*p* < 0.001, *p* = 0.002, respectively). Unfortunately, at the 2 h time point, there were too few pairs of urine samples to obtain a meaningful result (*n* = 3/23 in control mice in W1 and W7, and *n* = 19/71 in treated mice in W2, W4 and W5). The urinary excretion of iohexol (%) in female and male mice, measured seven times (at 2, 4, 6, 12, 15, 18 and 24 h) over 24 h after oral administration of iohexol, is presented in Figure 4C,D.

### 3.4. Correlation between the Urinary Excretion of Iohexol Measured at Seven Time Points and in Cumulative Urinary Samples over a 24 h Period in Mice

Table 2 presents the correlation coefficients between the urinary excretion of iohexol after the cumulative 24 h time point and the seven individual samples collected over 24 h in control and treated mice, including the differences between sexes (Appendix A). The sum of the urinary excretion of iohexol (%) from two up to three individual mouse urine samples produced a strong correlation that varied from 0.61–0.74, while from four up to six individual mouse urine samples combined produced a very strong correlation that varied from 0.84–0.88. Likewise, the correlation coefficients were strong between the urinary excretion of iohexol after the cumulative 24 h period and the individual mouse urine samples collected 2 h and 24 h after oral iohexol administration.

### 3.5. Correlation between the Histological Score for Duodenitis and the Weight of the Duodenum or Total Small Intestine

Histological duodenitis was assessed, as reported previously [38], also demonstrated in Appendix A. Ki–67–stained histological sections representing different duodenitis severity scores are demonstrated in Figure 5. The correlation between the histological score for duodenitis and the weight of the duodenum or the total small intestine is illustrated in Figure 6 and presented in Appendix A. A strong correlation coefficient was obtained between the histological score for duodenitis and the duodenum weight in female and male mice together and separately (Spearman’s rho: *n* = 0.76, *p* < 0.001, *n* = 27; 0.60, *p* = 0.020, *n* = 15; 0.71, *p* = 0.004, *n* = 15, respectively). The correlation between the histological score of duodenitis and the ratio of the duodenum or the total small intestine weight to animal body weight is presented in Appendix A. A strong correlation coefficient was obtained between the histological score of duodenitis and the ratio of duodenum weight to animal body weight in female and male mice together and separately (Spearman’s rho: *n* = 0.73, *p* < 0.001, *n* = 27; 0.70, *p* = 0.005, *n* = 15; 0.77, *p* = 0.001, *n* = 15, respectively). Data on the duodenum weight were obtained from a group of control mice (*n* = 3) for comparison.

### 3.6. Correlation between Urinary Excretion of Iohexol and the Histological Score for Duodenitis or the Weight of the Duodenum or Total Small Intestine

The correlation between the cumulative urinary excretion of iohexol in W4 and W5 and the histological score for duodenitis is presented in Figure 6. IP tests were performed in W4 and W5 of treatment, by which time the intestine is affected by T-cell-mediated inflammation in this model. The urinary excretion of iohexol in W4 and W5 is therefore more likely to reflect the histological assessment than W2. In treated male mice, there was a positive correlation between the urinary excretion of iohexol and the histological score for duodenitis in W4 and W5. This was most evident during the early time points after iohexol oral administration for the cumulative urine samples from 0–2, 0–4, 0–6, 0–12 and 0–15 h (Spearman’s rho: *n* = 0.83, *p* = 0.006, *n* = 10; 0.58, *p* = 0.003, *n* = 26; 0.55, *p* = 0.004, *n* = 26; 0.54, *p* = 0.005, *n* = 26; 0.58, *p* = 0.002, *n* = 26, respectively) (Appendix A) and was even more obvious in treated male mice in W5 (Figure 7 and Appendix A). There was also a positive correlation between the urinary excretion of iohexol and the duodenum weight for the cumulative samples from 0–2, 0–12, 0–15 and 0–18 h (Spearman’s rho: *n* = 0.77, *p* = 0.014, *n* = 10; 0.43, *p* = 0.030, *n* = 26; 0.41, *p* = 0.039, *n* = 26; and 0.42, *p* = 0.034, *n* = 26, respectively) (Appendix A) in treated male mice in W4 and W5 as well as in treated female mice in W4 for the cumulative 0–2 h urine sample (Spearman’s rho: *n* = 0.77, *p* = 0.036, *n* = 8). There was no correlation between the urinary excretion of iohexol and the total small intestinal weight (Appendix A) in either male or female treated mice.

## 4. Discussion

The findings from our study support the use of iohexol as a probe for IP testing in mice. They describe the time course of urinary excretion of iohexol during a 24 h period in both untreated mice and mice with intestinal inflammation induced by adoptive transfer of effector/memory T cells. In this mouse model, mice develop duodenitis, enteritis and colitis [31,37].

According to our results, the excretion of iohexol in urine peaks at 4 h after its oral administration in control and treated mice, followed by a lower peak at 24 h in control mice and only in Weeks 4 and 5 in treated mice. The number of urine samples at 2 h was too low to obtain a meaningful result. The profile of urinary excretion of iohexol was reminiscent of previous reports using ^51^Cr-EDTA as an IP marker in the serum of healthy dogs over a 6 h period [10]. IP varies along the axis of the gastrointestinal (GI) tract [45]. The iohexol IP peak at 4 h probably reflects small intestinal permeability to iohexol. Our results provide additional evidence that the permeation of iohexol across the intestine is greater in the first hours after its oral administration. This may support the use of a shorter mouse IP testing period in MCs than the common 24 h for the assessment of small intestinal permeability. A shorter testing period would increase the practicability of the method while allowing a reduction in the time spent by laboratory mice in the isolating MCs.

Single iohexol urinary samples produced moderate to strong correlation coefficients with the cumulative 24 h urinary excretion of iohexol. The highest correlation coefficients occurred when samples were taken at 2 or 24 h (r = 0.76 or r = 0.66, respectively). In control and treated mice of both sexes, the correlation coefficients were even stronger when comparing the cumulative 24 h urinary excretion of iohexol with the combination of single urine samples, specifically from three up to six single urine samples taken 2, 4, 6, 12, 15 and 18 h after iohexol administration (r = 0.74–0.88; *p* < 0.001). It is expected that the sum of multiple single iohexol urine samples better reflects the cumulative 24 h urinary excretion of iohexol than only the single urine samples. The latter are measured at one point in time, whereas the former more closely resemble the cumulative excretion of iohexol in urine.

One of the major disadvantages when using multiple collection time points in MCs is that not all individuals urinate at every required time point. In our study, 6 and 12 h were the time points with the highest number of animals urinating over the IP testing session. The cumulative urine volume within the first 6 h indicated that 95.8% of the animals delivered urine at least once during this period and 100% of mice within the first 12 h.

The use of the isolating MCs for prolonged periods (commonly 24 h) is mostly justified to allow the collection of sufficient volumes to run urinary IP tests. The volume of urine may influence the urinary excretion of the probes [46], and it should therefore be precisely collected and measured throughout the test. Our results demonstrated that urinary collection from the MCs over 6 or 12 h comprised more than half (59% or 68%, respectively) of the total median ± IQR excreted volume (0.27 or 0.31 mL out of the total 0.45 mL, respectively) over the 24 h IP test. The volume of urine that is needed to perform an enzyme-linked immunosorbent assay (ELISA) test normally varies from 0.010 to 0.020 mL. A 6 h MC urinary collection period may yield more than the required volume to perform the analysis in duplicate or even triplicate. It is known that in very small urine and blood samples from different species, the concentration of iohexol has been successfully analyzed by using ELISA [2,3,16,23]. A study from 2018 recorded a strong correlation between ELISA, high-performance liquid chromatography (HPLC) and neutron activation analysis (NAA) techniques when analyzing the iohexol concentration in canine plasma [2]. While these results cannot be directly extrapolated to mouse urine and blood samples, the same correlation between ELISA, HPLC and NAA is expected in mice.

The urinary excretion of iohexol was higher in treated than in control mice. The treated mice showed significant differences in iohexol excretion between W2 and both W4 and W5. This difference is consistent with the observation that the development of clinical intestinal disease in this model requires at least four weeks, following T cell reconstitution after homeostatic hyperproliferation of transferred T cells in lymphopenic Rag1-/- recipients, leading to body weight loss that usually starts between Weeks 4 and 5 [31]. Although these differences were observed in both sexes, male mice exhibited a greater increase in iohexol excretion between W2 and W4/W5 than females. In treated mice, iohexol excretion in urine in W4 or W5 was correlated both with the severity of histological duodenitis and weight of the duodenum or small intestine. All these correlations were more evident in W5 than W4 and were stronger in treated male than female mice. This was most evident during the early time points after the iohexol oral administration, with a very strong positive correlation at 0–2 h, a strong correlation at 0–4, 0–6 and 0–12 h and a moderate correlation at 0–15 h between the urinary excretion of iohexol and the histological score for duodenitis in treated mice in W5. In both female and male mice, there was also a strong positive correlation between the histological score for duodenitis and the duodenum weight and the ratio of duodenum weight to animal body weight. These findings suggest that the use of iohexol for measuring increased permeability as a result of intestinal inflammation in mice is feasible.

Under visual examination, our mice from both groups (control and treated) were in good physical condition and behaved normally over the length of the study. Only one male mouse from the treated group was returned to its home cage during the ongoing IP test in W5. This mouse was displaying changes in behavior, i.e., apathy and rough hair coat, probably related to treatment. We did not include an ethogram or measurement of any of the stress markers in mouse blood or urine, e.g., corticosterone [47].

In general, control female and male mice and treated female mice gained weight over the length of the study to a lesser or greater degree, while treated male mice apparently lost weight. This difference could reflect a treatment-related effect in the treated male mice. However, the total weight loss in all animals did not even reach 5%, so the weight was still within normal value ranges [48,49].

Stool consistency and appearance for control and treated mice were mostly graded as normal. Occasional transient wet or pasty stools were observed in W4 or W5, respectively, in some female and male mice from the treatment group. These changes in stool consistency were only obvious in visual observation during MC housing and were not detected during the weekly body weight measurement sessions. One urine sample was discarded from iohexol analysis due to fecal contamination. Food contamination was also found in some of the urine sample containers. This could not have influenced the iohexol analysis since the marker was applied via gastric intubation and not with the food.

The use of 24 h MC housing is the current gold standard technique for the collection of mouse urine samples. This method is, however, susceptible to different contaminants such as stools or food, which may affect the results to an unknown greater or lesser degree. Alternatives to MCs such as hydrophobic sand [50,51,52] are refined techniques that have not been yet extensively used in rodents, unlike MCs. Efforts should be made to promote the implementation of easy-to-use, noninvasive, less stressful and less variable refined devices or methods for the collection of urine samples in rodents during IP testing.

To the best of our knowledge, the performance of a urinary iohexol IP test has never been assessed in mice. Our main aim was to analyze the iohexol IP test method in an inflammatory intestinal disease model. The results confirmed our hypothesis that mice suffering from intestinal inflammation display increased iohexol IP. We also report a time course of iohexol urinary excretion. In the future, it may be possible to assess IP in a period shorter than 24 h, specifically within the first 6 or 12 h after oral iohexol administration.

Single housing in metabolic cages on a wire-mesh floor deprives the animals of cage enrichment and bedding, living space and group housing. Rodents have been shown to only acclimatize to MCs after three to four days [29,30]. All these factors are known to compromise animal welfare, which may induce stress or discomfort in laboratory rodents [26,30,53,54], and thus may confound the research data. Many other factors, such as age, strain, sex, social position and housing and husbandry regimens, might also influence research data [55]. According to the literature, prolonged stress periods are considered to have significant consequences for animal health and welfare by changing the physiological parameters of the reproductive, endocrine, cardiovascular, digestive and immune systems [56,57,58,59]. Rodents can undergo stress-induced changes in the intestine and are therefore even suggested as model animals for irritable bowel syndrome [59]. When studying the influence of inflammation on IP, it is important to keep the stress level to a minimum to reduce its influence on intestinal function as a confounding factor in gastrointestinal research.

Murine models to replicate animals and human diseases are still needed to provide basic information regarding intestinal and no intestinal diseases in humans, in vivo research models or veterinary patients. These murine models have notable advantages in terms of a large available sample size and a well-understood, genetically homogeneous background [60]. Laboratory mice could directly benefit from the improvement and standardization of iohexol IP testing protocols.

In future studies, the excretion of iohexol in serum and plasma after oral ingestion of the probe molecule may be compared with the urinary excretion of iohexol in mice. Serial blood sampling in mice during IP tests is still impractical, mainly because of the invasiveness of the technique. The information gained from the most suitable urine sampling time point(s) can inform the methodology of blood sampling to assess iohexol IP in mice. In future research, iohexol IP tests in rodents may be equally performed on either urine or blood. The use of blood samples in small amounts could replace the use of MCs, and urine collection may no longer be required for IP tests in mice.

## 5. Conclusions

In summary, we report on the performance of a refined permeability testing protocol using the urinary excretion of iohexol in mice. Iohexol excretion in urine in Weeks 4 or 5 of treatment was correlated with increased severity of histological duodenitis, suggesting that iohexol may be used as a marker to assess intestinal damage in this inflammatory disease model. We also addressed the use of MCs to collect urine in mice and obtained a time course of the urinary excretion of iohexol over 24 h. Urine collection within 6 h after iohexol oral gavage may be sufficient to evaluate IP in a mouse model of intestinal inflammation, although further studies are still warranted to validate the findings. Those studies may confirm the value of iohexol as a diagnostic tool for the assessment of IP in rodent species in research and clinical settings.

## Figures and Tables

**Figure 1 animals-11-00079-f001:**
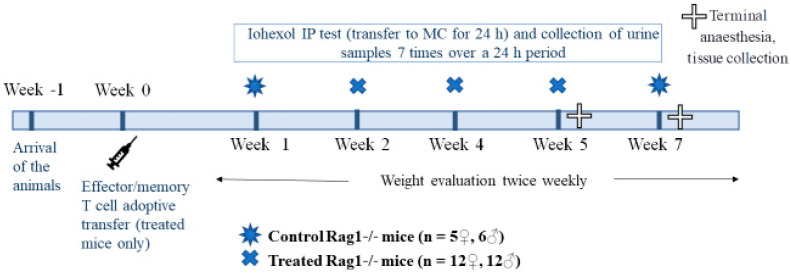
Study design. IP, intestinal permeability; MC, metabolic cage.

**Figure 2 animals-11-00079-f002:**
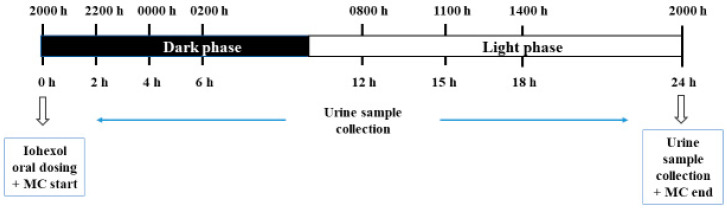
Timeline for the metabolic cage housing and urine sampling. MC, metabolic cage.

**Figure 3 animals-11-00079-f003:**
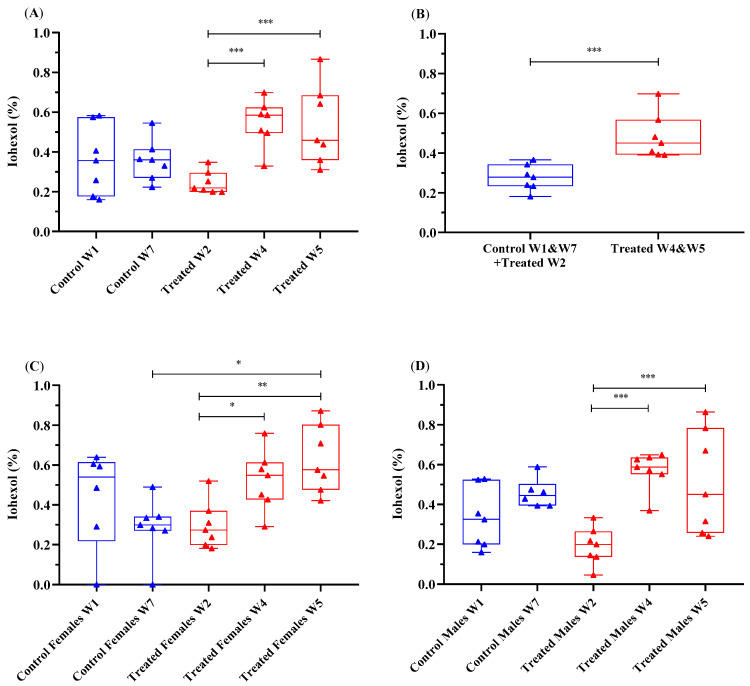
Urinary excretion of iohexol (%) during 24 h following oral administration in all mice (**A**,**B**) and (**C**) female and (**D**) male mice over 24 h, with data combined for the 7 weeks of intestinal permeability testing. Data are presented on a base-10 log scale (Log10 (1 + x)). The horizontal line in the box is the median (50% percentile), and the upper and lower limits of the box indicate the 75% upper and 25% lower quartiles, respectively. The limits of the upper and lower vertical lines represent the maximum and minimum data results, respectively. Dots represent each of the seven urinary collection time points during the 24 h period. * *p* < 0.05, ** *p* < 0.01, *** *p* < 0.001.

**Figure 4 animals-11-00079-f004:**
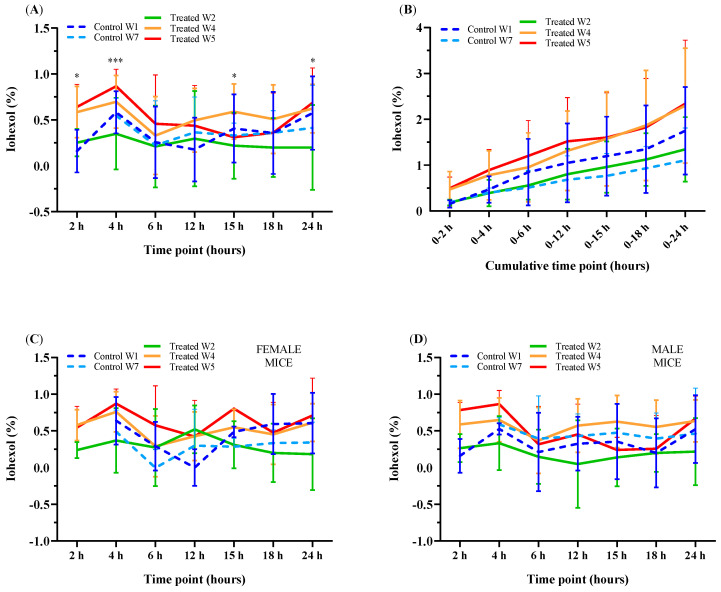
Urinary excretion of iohexol (%) in control and treated (**A**) female (**C**) and male (**D**) mice at seven separate urine collection time points or (**B**) at seven cumulative time points (after 2, 4, 6, 12, 15, 18 and 24 h) over 24 h after oral administration of iohexol. Data are shown on a base-10 log scale (Log10 (1 + x)) and expressed as means ± SD. * *p* < 0.05, *** *p* < 0.001.

**Figure 5 animals-11-00079-f005:**
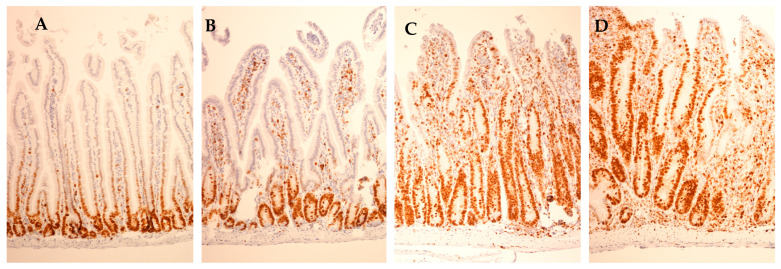
Histological sections of duodenum stained for Ki-67, representative of different duodenitis severity scores. (**A**) Histological duodenitis score 0, (**B**) score 3, (**C**) score 6 and (**D**) score 8. Proliferating epithelial cells and other dividing cells within the lamina propria stained brown (original magnification 100×). Note the extension of the proliferating crypt epithelial zone along the crypt/villus axis with increasing duodenitis scores.

**Figure 6 animals-11-00079-f006:**
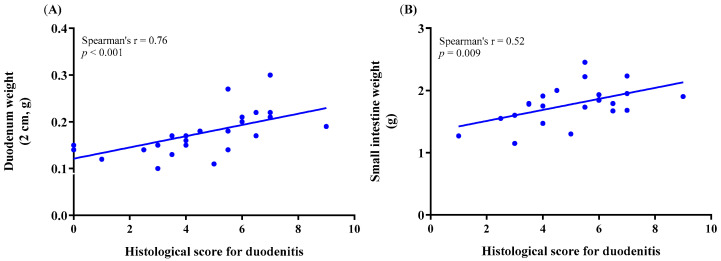
Spearman’s rank correlation coefficient between the histological score for duodenitis and the weight of (**A**) the duodenum and (**B**) the total small intestine in treated and control mice combined.

**Figure 7 animals-11-00079-f007:**
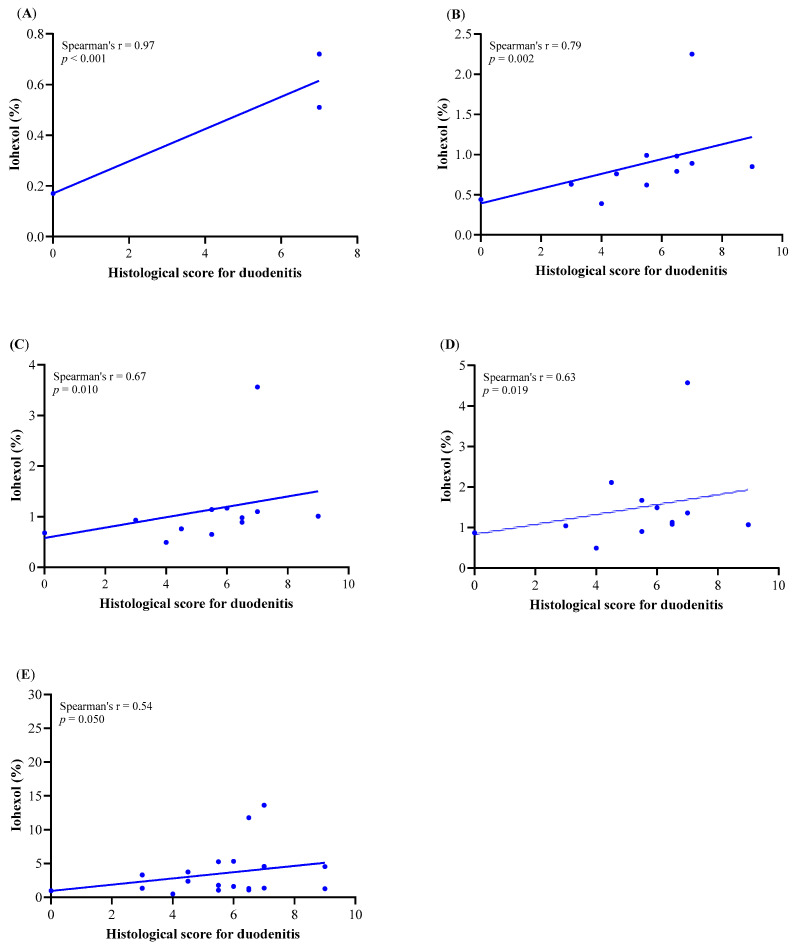
Spearman’s rank correlation coefficient between the histological score for duodenitis and the urinary excretion of iohexol in time periods (**A**) 0–2 h, (**B**) 0–4 h, (**C**) 0–6 h, (**D**) 0–12 h and (**E**) 0–15 in treated male mice in Week 5.

**Table 1 animals-11-00079-t001:** Urinary excretion of iohexol (%) in control and treated mice measured seven times over a 24 h period after iohexol oral administration.

Time Point (Hours)	Control W1, Iohexol (%) ^1^	Control W7, Iohexol (%) ^1^	Treated W2, Iohexol (%) ^1^	Treated W4, Iohexol (%) ^1^	Treated W5, Iohexol (%) ^1^	*p*-Value, One-Way ANOVA ^2^
2	0.16 ± 0.23	0.27 ± 0.00	0.25 ± 0.15	0.59 ± 0.28	0.64 ± 0.25	* 0.024
4	0.58 ± 0.23	0.55 ± 0.19	0.35 ± 0.39	0.70 ± 0.29	0.87 ± 0.18	***
6	0.26 ± 0.39	0.22 ± 0.49	0.21 ± 0.45	0.33 ± 0.42	0.46 ± 0.53	0.505
12	0.18 ± 0.35	0.36 ± 0.38	0.30 ± 0.52	0.50 ± 0.35	0.44 ± 0.44	0.276
15	0.41 ± 0.37	0.33 ± 0.14	0.22 ± 0.36	0.59 ± 0.30	0.31 ± 0.26	* 0.047
18	0.36 ± 0.45	0.36 ± 0.24	0.20 ± 0.32	0.51 ± 0.37	0.36 ± 0.43	0.204
24	0.58 ± 0.40	0.41 ± 0.46	0.20 ± 0.46	0.62 ± 0.26	0.68 ± 0.38	** 0.002

^1^ Results are expressed as means ± SD of the percentage of iohexol on a base-10 log scale (Log10 (1 + x)). ^2^ Statistically significant difference between groups, * *p* < 0.05, ** *p* < 0.01, *** *p* < 0.001.

**Table 2 animals-11-00079-t002:** Correlation coefficients between the urinary excretion of iohexol in both the seven individual and the cumulative urinary collection samples, collected during 24 h after the oral administration of iohexol to control (W1 and W7; total *n* = 22) and treated (W2, W4 and W5; total *n* = 71) female and male mice.

ALL MICE	IOHEXOL (%)											
	Single Samples							Two Samples	Three Samples	Four Samples	Five Samples	Six Samples
Iohexol (%) (0–24 h)	2 h	4 h	6 h	12 h	15 h	18 h	24 h	2 + 4 h	2 + 4 + 6 h	2 + 4 + 6 + 12 h	2 + 4 + 6 + 12 + 15 h	2 + 3 + 4 + 6 + 12 + 15 + 18 h
Correlation coefficient	0.76	0.58	0.48	0.54	0.55	0.59	0.66	0.61	0.74	0.84	0.84	0.88
n, number of mice	22	73	77	77	47	63	73	74	91	93	93	93
*p*-value	***	***	***	***	***	***	***	***	***	***	***	***

Correlations were calculated using Spearman’s rank coefficients where appropriate. Significance levels are expressed as: *** *p* < 0.001.

## Data Availability

Not applicable.

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
