# Peer review of "Urinary Excretion of Iohexol as a Permeability Marker in a Mouse Model of Intestinal Inflammation: Time Course, Performance and Welfare Considerations"

_animals, 2021, doi:10.3390/ani11010079_

Round 1

Reviewer 1 Report

This is an interesting study that could significantly shorten the amount of time animals need to be single housed for excretion studies using Iohexol, representing a nice refinement technique. In general the manuscript is well written but more detail is needed in multiple sections.

  1. section 2.1
    1. are the wild type animals purchased directly from Jax and C57BL/6J or are they bred in house or possibly heterzygotes? how old were these mouse at time of cell harvest? male or female?
    2. please include additional information about the 'similar diet' - was this made in house or contracted to a company produce? was the diet irradiated or autoclaved prior to offering it to the mice? was the quantity of gluten tested in the diet after processing to verify the amount available to the animals to ingest?
    3. How many nestlets were provided?
    4. how much Aspen would wool was provided?
    5. how many animals were housed per cage?
    6. were animals kept in same sex groups?
    7. were animals individually identified with tattoos or ear tags?
  2. section 2.2.1
    1. can you please clarify what you mean by 'treatment of mice for the induction of intestinal inflammation'? Are these the splenic memory T cells?  Were these administered to all of the Rag-/- mice, both control and treated? Was control versus treated the injection of the splenic memory cells or the diet fed?  
    2. It is unclear what is meant by the control and treatment groups - Is the difference the diet or the cell treatment?
    3. When did the animals start on their respective dietary treatments? Before intestinal inflammation induction? What is the function of the diet? who was fed what?
    4. How and when were animals exanguinated? Were all animals kept until week 7?
    5. What does and route of ketamine-xylazine was used?
    6. Why do you use different time points for the control versus treated animals?
  3.  section 2.2.2
    1. were donor cells tested for rodent pathogens or assumed to be clean because they were from the facility?
    2. What was the volume and carrier solution used for the IP injection of splenic memory T cells?
  4. section 2.2.5
    1. How and when were animals exanguinated? where all animals kept until week 7
    2. What does and route of ketamine-xylazine was used?
    3. please include a figure with representative images of the scoring systems used.
  5. section 3.1
    1. sentences that start and end ' even though the mice did not exhibit overt signs....heads tucked under their chests' - I don't doubt your observational finding but it is very subjective and biased. I recommend removing this because it detracts from the otherwise scientific nature of the study.  In the future, a better approach to collect his sort of data would be through the use of a scoring system that you implement both while housed in standard caging and in metabolic caging.
    2. was the male mouse, that was removed in week 5, included in all other data analysis? was he euthanized in week 5 or kept until week 7?
    3. I am very confused about your n for each of the time points.  Are all 5 female and 6 male control mice tested at weeks 1 and 7 and all 12 female and 12 male treated mice tested at weeks 2, 4, and 5?
    4. If the animals were randomly assigned to groups, why is there such a marked difference in body weights between control and treated males at all time points, especially prior to induction of disease? Why does panel A of Figure s1 stop at week 5 if the study was 7 weeks? Also, the treated male group stops at 4.5 weeks which does not make sense. It seems as if some data is missing or something was not fulling described in the methods.
  6. section 4
    1. the manuscript points out the short comings and welfare issues of metabolic cages, but doesn't not discuss the potential for the use of hydrophobic sand as an alternative to further improve welfare.  There are a number of relevant publications that could be used to include this as a valid point in welfare considerations for these types of studies. References include: Hoffman et al., 2018 JAALAS; Hoffman et al. 2019 Molecular genetics and metabolism reports; Reverte et al. 2020 Journal of diabetes and its complications; Vaara et el. 2020 Antibiotics; Shubitowski et al. 2019 Physiological Reports.

Reviewer 2 Report

The manuscript “Urinary Excretion of Iohexol as a Permeability Marker in a Mouse Model of Intestinal Inflammation: Time Course, Performance and Welfare Considerations” it is a very interesting study that allows to significantly reduce the time of the tests in which the animals need to be in the metabolic cages, in Iohexol excretion studies, thus reducing the associated stress and discomfort, representing a good example of refinement.

In general, the manuscript is well written, and is quite balanced in all its sections. I believe that the publication of this work in this Special issue of Animals is justified. However, there are some aspects that I would like to clarify with the authors and that need to be addressed before the manuscript can be published. 

SIMPLE SUMMARY AND THE ABSTRACT

The simple summary and the abstract are very assertive, well written and targeted, allowing us to immediately understand the objectives of the work, giving a perfect framework.

1. INTRODUCTION

Very good, well written and with the necessary information to frame the study and its objectives.

2. MATERIALS AND METHODS

Comment 1: Unless they are Journal rules, I think that section “2.3. Study approval” should be the first one, ie 2.1., or even just after the general title of the section. 

2.1. Animals and Housing

Comment 2: It was not clear to me whether there was a previous period of acclimatization of the animals and if so, how long it will take to start the test with the administration of T cells. If so, in addition to referring to the text, perhaps in figure 1 between the injection administration and the start of the experiment (see Comment 4)

Comment 3: The control group only distinguishes itself from the experimental group because it does not receive T cells?

2.2.2. Induction of gastrointestinal inflammation

Comment 4: I suggest writing “Gastrointestinal inflammation was induced in RAG1-/- recipients by intraperitoneal injection (Figure 1)…”and the days in the figure 1 if necessary write before the start of the experiment (see Comment 2)

Comment 5: What is the vehicle solution and its volume to administer T cells?

2.2.5. Tissue collection and histological score for duodenitis

Comment 6: Intestine weight refers to intestines without food content?

Comment 7: It would not be more suitable to correlate the histological score for duodenitis with the ratio’s intestinal weight/animal weight?

Comment 8: Regarding immunohistochemistry to assess Ki67 expression:

  • What are the incubation period and temperature?
  • HIER was needed?
  • What controls are used?
  • What are the criteria for quantifying Ki67 positive cells?

Comment 9: I think it would be interesting to include a table with the scoring systems used.

Comment 10: I believe that the histological criteria should be better explained, the Villus/crypt architecture (V/C ratio, height?), how villus lamina propria and crypt are evaluated and the diameter measured (with Computer-assisted morphometric measurements?), how many villi and crypts are evaluated for diameter, with what magnification, means of how many independent measurements etc.

It is a fact that the authors refer to reference 44, which I read, but in this refer to the Reference 18, which I also read (both with very beautiful photomicrographs, mainly at 18), and in this refer to the references 14 and 15, which I also read. Wouldn't it be better to elucidate the reader soon, in the manuscript itsel, about the histomorphometric criteria?

3. RESULTS

Comment 11: Please explain this sentence there seems to be some contradiction between “excessive excitement“ and “apathy”.  “This animal displayed excessive excitement and a change in behaviour, i.e. apathy and rough hair coat, at the start of the experiment, and was excluded from iohexol analysis”.

Comment 12: Although the authors mentioned in the Materials and Methods the realization of immunohistochemistry for Ki67, there are no results for this proliferation marker.

4. DISCUSSION

The discussion seems to be good with actual bibliography and the authors assume that some situations may raise some doubts for example “All these factors are known to compromise animal welfare, which may induce stress or discomfort in laboratory rodents [26,30,52,53], and thus may confound the research data. Many other factors, such as age, strain, sex, social position and housing and husbandry regimens, might also influence research data”.
